# Historical evolution of the geomagnetic declination at the Royal Observatory of Madrid.

Jose Manuel Tordesillas[1,2], Francisco Javier Pavón-Carrasco[3,4], Alberto Nuñez[1], Ana Belén Anquela[2]

[1]Instituto Geográfico Nacional, Madrid, 28003, Spain
[2]Universitat Politècnica de València, Valencia, 46022, Spain
[3]Departamento de Física de la Tierra y Astrofísica, Universidad Complutense de Madrid, Madrid, 28040, Spain
[4]Instituto de Geociencias (CSIC-UCM), Madrid, 28040, Spain

*Correspondence to*: Jose Manuel Tordesillas (jmtordesillas@transportes.gob.es)

**Abstract.**

The agonic line, which represents geomagnetic declinations of 0º, recently crossed the Royal Observatory of Madrid (ROM) in December 2021, causing a shift in declination values from west to east. This event constitutes a notable milestone for this significant observatory, where the first geomagnetic observation series commenced around 1855 in Spain. In this work, taking advantage of the occurrence of this event, a detailed study has been conducted to investigate the historical evolution of the magnetic declination at ROM to decipher prior occurrences of the agonic line crossing this place. Despite the ROM hosted the first series of geomagnetic measurements in Spain, the present lack of geomagnetic measurements in this observatory makes necessary to extend the declination measurements to other observatories distributed throughout the Iberian Peninsula to better define the passage of the agonic line since 1855 up to the present. For epochs prior to 1855, a bibliographic search for declination measurements conducted in the Iberian Peninsula has been carried out, complemented by historical data from the HISTMAG database. As a result, a time-continuous curve of geomagnetic declination is generated from 1590 to 2022 at the ROM coordinates. The declination curve reveals that the agonic line also crossed the ROM 400 years ago (around 1600) passing from west to east declination values.

## 1 Introduction

At the end of 2021, the agonic line (magnetic declination line with 0º values) crossed the Royal Observatory of Madrid (ROM) changing the declination on this place from west values to east values. According to the Geomagnetic Reference Model for the Iberian Peninsula and Balearic Islands (also named as Geomagnetic Iberian Model, Puente-Borque et al., 2023; more information in S1 of the Supplementary Material) this event occurred on 12 September 2021 (see Fig. 1). The interest on this event, considering that ROM does not have a long tradition in geomagnetism, and it was never equipped with variometers for continuous recording, comes from the fact that the first regular declination observations in Spain were made in this place.

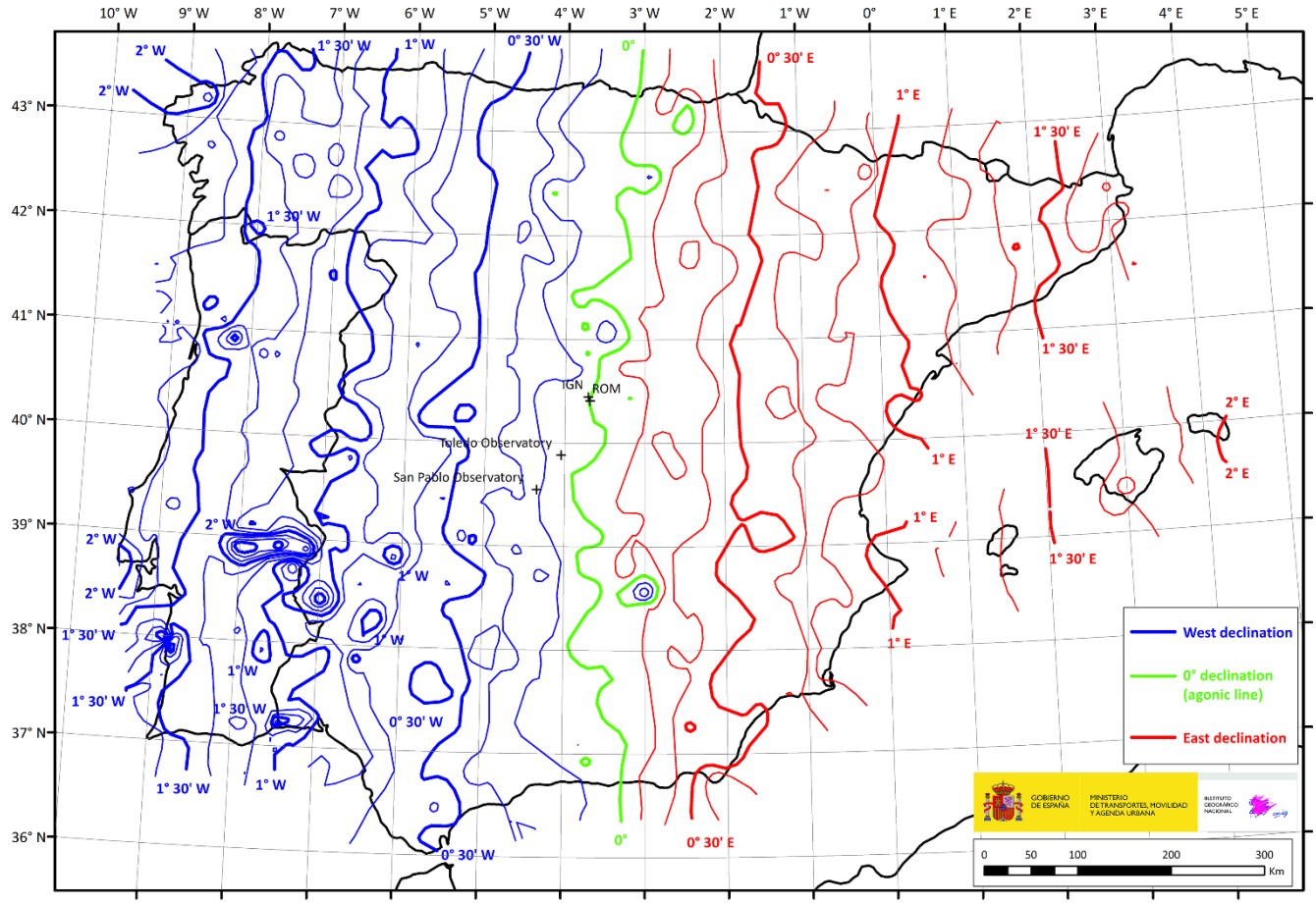

30

**Figure 1: Declination map of the Iberian Peninsula for September 12, 2021 according to the Geomagnetic Reference Model for the Iberian Peninsula and Balearic Islands.**

The event was monitored by Instituto Geográfico Nacional (IGN) showing in near real time the declination deduced for the Royal Observatory of Madrid between 2021 and 2023 (see S2, Supplementary Material). To get the real time declination at ROM, we transfer there the declination data observed at San Pablo de los Montes Observatory (SPT), the closest Spanish observatory (110 km far away from ROM). The spatial transference of the declination data from SPT site to ROM coordinates was carried out using the spatial gradient provided by the Geomagnetic Iberian model. Original daily mean declination data from SPT and the data transferred to ROM are plotted in Fig. 2a for the period 1 January 2021 to 1 January 2023. The transferred data indicate that the agonic line crossed the ROM around December 06, 2021. Note that the difference between the date given in Fig. 1 (September 12, 2021) and its equivalent of Fig. 2a (December 06, 2021) is due to the magnetic anomalies beneath both ROM and STP observatories (the so-called anomaly biases) that slightly perturb the declination values given by the main geomagnetic field.

43

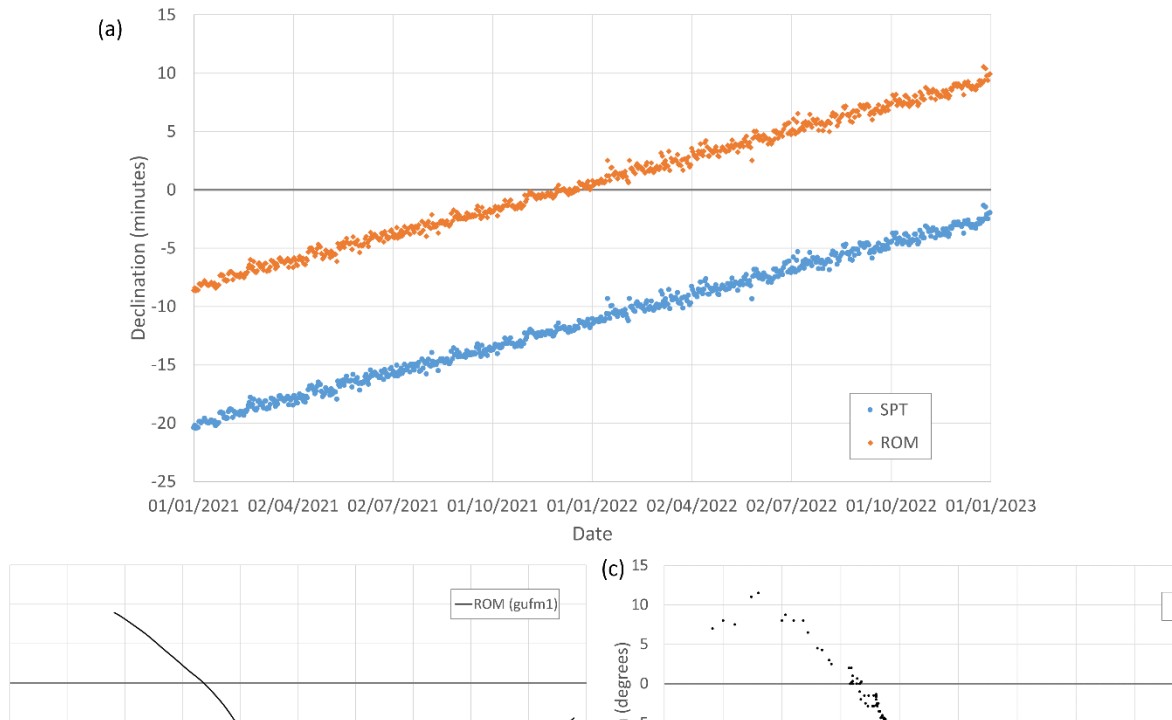

Figure 2: (a) Daily mean declination data recorded at SPT observatory and the transferred declination data at ROM observatory. Declination data is transferred from SPT to ROM by using the spatial declination gradient derived from the Geomagnetic Iberian Model. (b) Annual mean declination values at ROM estimated from the *gufm1* model. (c) Historical records of declination in Paris.

Here, we also focus our analysis in previous crossings of the agonic line at ROM during the historical period covered by instrumental geomagnetic data, i.e. the last four centuries. At first glance, and according to the historical geomagnetic reconstruction *gufm1* based on a complete compilation of historical observations, mainly taken in naval shipping (Jackson et al., 2000), it seems that the last time that this event occurred was around 1668 (Fig. 2b). This epoch is in agreement with the declination data recorded in other French geomagnetic observatories (Alexandrescu el al., 1996; Mandea and Le Mouël, 2016) close to Spain (Fig. 2c). This previous crossing of the agonic line was characterized by an eastward drift, i.e. the declination changed from east to west values.

Summarising, the goal of this work is to highlight the historical significance of the Royal Observatory of Madrid, which served as the first site for geomagnetic measurements in Spain. Additionally, we have compiled a comprehensive dataset of Spanish geomagnetic declinations derived from a variety of sources and spanning the last four centuries. Then, we have

transferred all the declination data to ROM coordinates to develop a time-continuous declination curve that allows
determining the epochs at which the agonic line crossed the ROM observatory during the last centuries.
**2 The Royal Observatory of Madrid**
In 1785, King Carlos III of Spain decided to establish an Astronomical Observatory in Madrid and commissioned its design
to the renowned architect Juan de Villanueva (Tinoco, 1951). Its construction began around 1790 near the Buen Retiro
Palace. Concurrently, experts were recruited to work at the Observatory, and a collection of instruments was acquired.
However, just as the works were completed, the Napoleonic invasion of Spain in 1808 led to the destruction of
documentation and instrumentation, resulting in significant damage to the Observatory building, which was abandoned for
years. Reconstruction began in 1846, including the training of new staff and the acquisition of new instruments. By 1851, the
Royal Observatory of Madrid was  operational (Fig. 3 shows a picture of the Observatory taken in 1853).

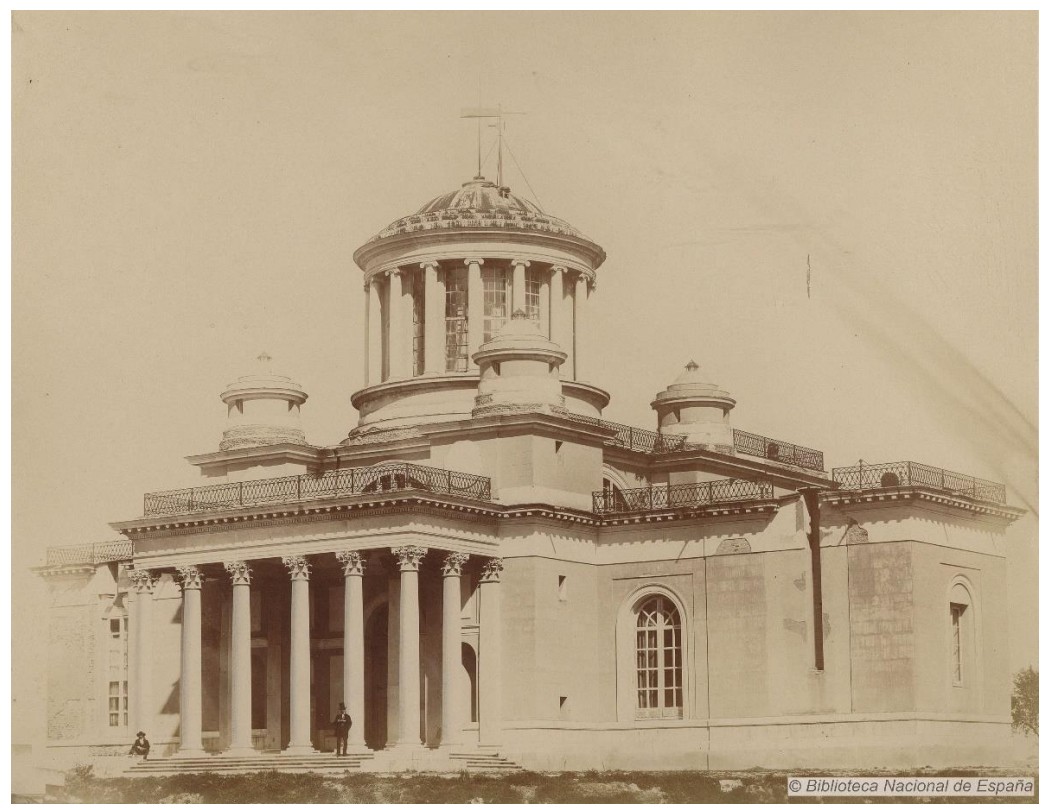


**Figure 3: The Royal Observatory of Madrid in 1853 (Source: Biblioteca Nacional de España,**
**http://bdh.bne.es/bnesearch/detalle/bdh0000027343)**

In addition to the astronomical section, the new Observatory incorporated a meteorological division, acquiring the first
geomagnetic instruments in 1853 (Real Observatorio de Madrid, 1867): a) two magnetometers, to measure the horizontal and
vertical forces, with their corresponding telescopes. b) One *Barrow* theodolite, to determine the magnetic declination. c) One
*Barrow* dip circle. d) Two magnetized bars with their armours (see Table 1). These instruments were operated by Mr. Rico
Sinobas, the responsible of the meteorological observations, performing the first series of geomagnetic declination and
inclination measurements along the month of September of 1855. This constituted the first continuous time series of
geomagnetic observations made in a location of the Iberian Peninsula (Rico Sinobas, 1856; see also Tables S1 and S2 and
Fig. S1 and S2 of the Supplementary Material).
The declination series of observations were adjusted to the recommendations of relevant contemporary magnetic
observatories, referring the time to that given by the Observatory of Gottingen (Germany) and measuring during the hours of
maximum and minimum variation. Two daily declination measurements were observed at 2h 30m and at 20h 00m (it seems
that the time recorded here is the astronomical time and it is needed to add 12 hours to get the Universal Time). Meanwhile,
inclination measurements (only 7 inclination measurements were observed along the month) are consigned to be made at 9h
00m (in the morning) or at 15h 00m (in the afternoon). We have digitized the magnetic declination data obtained by Rico
Sinobas and the daily mean values are plotted in Fig. 4.

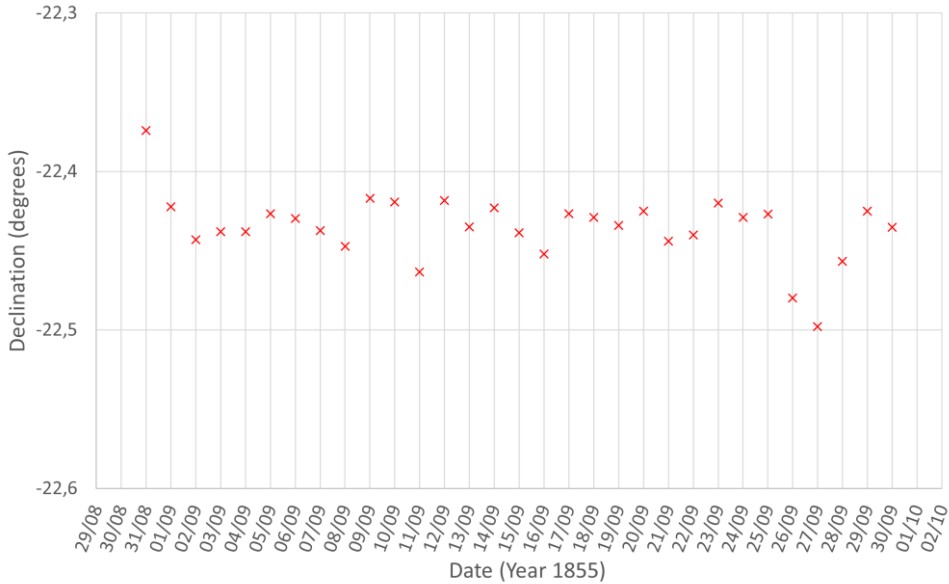


**Figure 4: Observed declination data by Rico Sinobas at the Royal Observatory of Madrid.**
To evaluate the Sinobas' declination data, we have compared them with the declination given by the historical geomagnetic
model *gufm1* at the same period and coordinates. The *gufm1* model provides a value of 20º 19' W, about 2º of difference
with the Sinobas' series. This difference could be due to the anomaly bias that characterized the crustal field beneath ROM
(this anomaly bias is not considered in the *gufm1* model, that only provides the main geomagnetic field). However, a
difference of 2º is too large to be considered of crustal origin. In fact, this issue was already highlighted by De Prado (1858)
after reviewing the declination values obtained by Dr. Lamont during his campaign in Spain to create an European magnetic
chart (Lamont, 1858). The value measured by Dr. Lamont for Madrid on 1st July 1857, was 20º 12' west, which closely
aligns with the *gumf1* model predictions. As a possible explanation, it was supposed that the measurements made by Rico
Sinobas were influenced by the large masses of iron used in the construction of the Observatory building. Although this
constant local influence seems not to affect to the recorded time variability in declinations (with a maximum difference of
about 13.5' between maximum and minimum values), this set of data is not useful for the purpose of our analysis.
Regarding the rest of geomagnetic instruments at the ROM, no measurements made with the magnetometers of H and Z have
been found. These instruments are missing with exception of the Barrow theodolite (see Fig. 5a) that is still preserved and
exhibited at the ROM and detailed in Instituto Geográfico Nacional (2012).
In 1878, a *Brunner* theodolite and a *Brunner* inclinometer were acquired (Fig. 5b,c). These instruments were installed as far
as possible from any potential sources of disturbance that could distort the measurements. One year later, in 1879, regular
observations of magnetic declination and inclination began at the ROM (Real Observatorio de Madrid, 1890). The
inclinometer broke down in 1892. In 1900, a new collection of magnetic instruments was acquired, consisting of a *Brunner*
theodolite and a *Brunner* inclinometer (Fig. 5d) manufactured by the company Salmoiraghi, Milano (Batlló, 2005; Instituto
Geográfico Nacional, 2012). These instruments are summarized in Table 1, and most of them can be visited in the ROM's
exhibition hall of historical instruments.

Table 1. Geomagnetic instrumentation

| Name | Period | Component | Sensitivity |
|---|---|---|---|
| Magnetic theodolite *Barrow* | 1853-? | D | 1' |
| Dip circle *Barrow* | 1853-? | I | Unknown |
| Horizontal magnetometer | Use unknown | H | Unknown |
| Vertical magnetometer | Use unknown | Z | Unknown |
| Magnetic theodolite *Brunner* | 1879-1900 | D | 1' |
| Inclinometer *Brunner* | 1879-1892 | I | 1' |
| Magnetic theodolite *Brunner* | 1900-1901 | D, H | 1', 10 nT |
| Inclinometer *Brunner* | 1900-1901 | I | 1' |





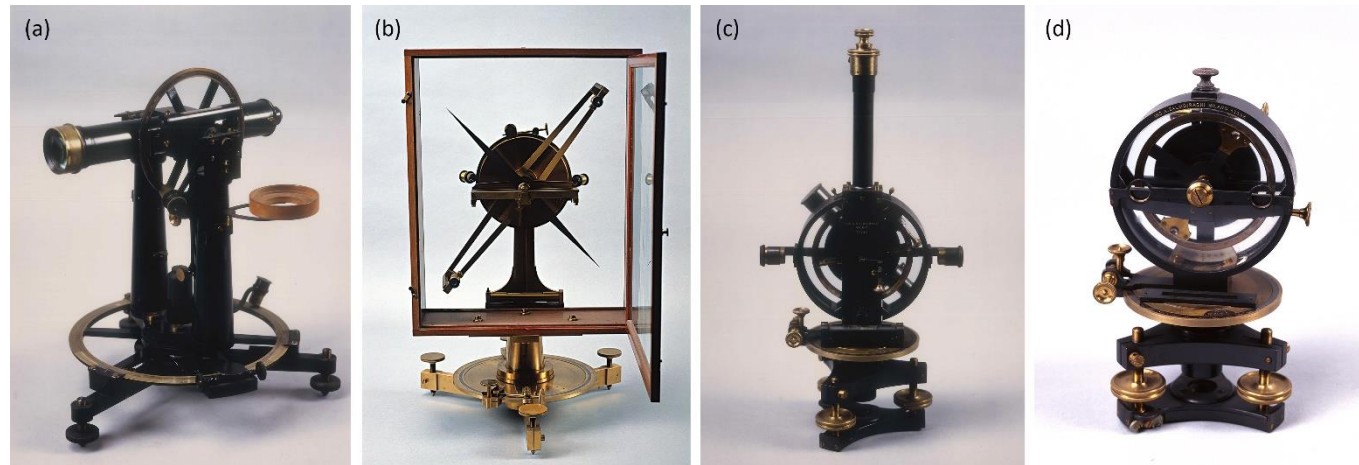

**Figure 5: Magnetic instruments used in the Royal Observatory of Madrid (source: IGN archive, © Instituto Geográfico Nacional, CC-BY 4.0 ign.es): (a) Magnetic theodolite *Barrow* (1853), (b) Inclinometer *Brunner* (1879), (c) Magnetic theodolite *Brunner* (1900), (d) Inclinometer *Brunner* (1900).**

The ROM geomagnetic observations were carried out between 1879 and 1901, published in the historical yearbooks from 1890 to 1904, and they were interrupted since 1902 due to the increase in electrical installations near the Observatory (Instituto Geográfico y Catastral, 1933). Declination measurements were made every day at 08:00 and 13:30 (local time), close to the maximum and minimum daily value of this element. Unfortunately, only mean values for every decade of days and their average were published. Pro et al. (2018) digitized these declination data and compared them with the *gufm1* model providing a good agreement between data and model with better stability over the years and increasing differences since 1897 (see Fig. 6a).

The series measured by Rico Sinobas during September 1855, and other previous declination values for the city of Madrid that were noted by him (Rico Sinobas, 1856) are also shown in Fig. 6b (the full dataset recompiled by Rico Sinobas is given by Fig. S3 of the Supplementary Material). We have also estimated the declination values for these epochs using the *gufm1* model (see Fig. 6b). Results show discrepancies between the Spanish declination measurements and the model predictions that increase for epochs before 1880. After 1904, the ROM was integrated in the IGN, and no further magnetic measurements were conducted at this location.

As it can be seen in Fig. 6, the amount of declination data available for the coordinates of the ROM is very scarce and it is impossible to define a declination curve using only these data covering the last centuries. In the following section, we present other source of data that will help to solve this problem.

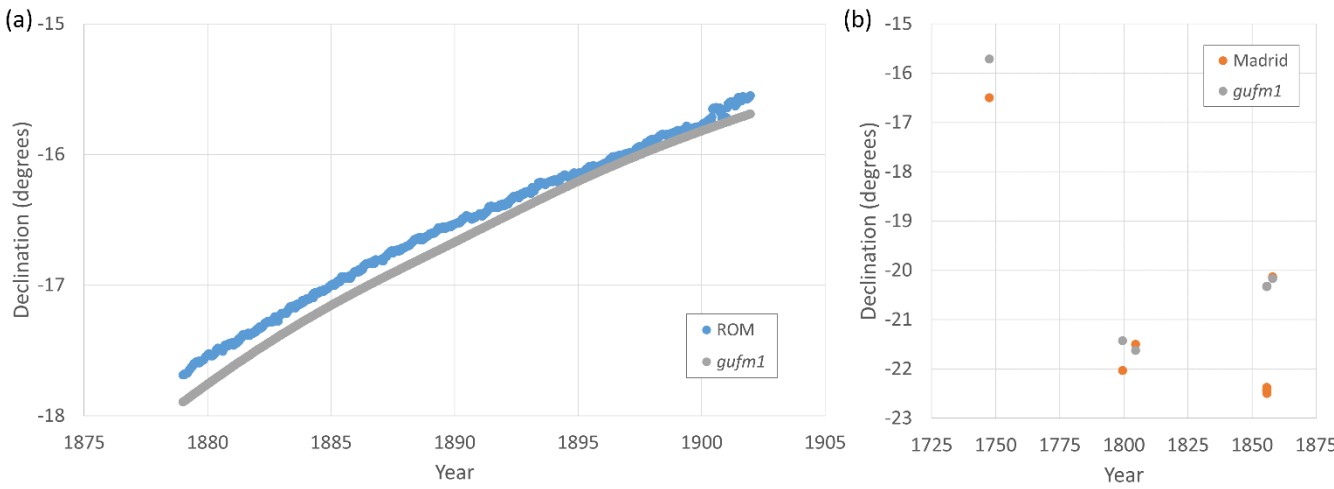


**Figure 6: (a) Declination values measured at ROM and estimated from the *gufm1* model in the period 1879-1901. (b) Declination**
**values in Madrid noted by Rico Sinobas (1856) and their corresponding estimations from the *gufm1* model.**
**3 Other Spanish observatories**
The Royal Observatory of Madrid was the first observatory in Spain to take regular measurements of the magnetic field as
part of the meteorological observations. Unfortunately, it was not a specific geomagnetic observatory with continuous
recording of the magnetic field. However, in Spain, a network of geomagnetic observatories has been in operation since the
late 19th century, with their numbers steadily growing throughout the 20th century (see their locations in Fig. 7). Many of
these observatories continue to function to this day. Here we provide a brief summary of their past history.
**San Fernando Observatory (SFS).** The Spanish Navy installed this geomagnetic observatory as a part of the Astronomical
Observatory of San Fernando (San Fernando, Cádiz). As well as at the ROM, regular geomagnetic observations were started
in 1879 (see Fig. 8a), but with more facilities: one independent pavilion constructed without magnetic elements, isolated and
buried, where the magnetometers were installed (Azpiazu and Gil, 1919).
It was equipped with a set of magnetographs *Adié* that continuously recorded the variations of the geomagnetic field. In
addition, a magnetometer *Elliot* and an inclinometer *Dover* were available to make absolute measurements. The recorded
data from SFS observatory have been published without interruption in the Observatory's yearbooks from 1891 until now.
In the decade of the 1970's the railway electrification in the line Cádiz-Sevilla caused significant interferences over the
geomagnetic records. For that reason, the geomagnetic observatory was moved to a new location, 8 km far at NE of the
original location, in Puerto Real (Cádiz). It was operative from 1978 until 2004 (Real Instituto y Observatorio de la Armada
en San Fernando, 2021). However, after detecting new interferences in the geomagnetic records, it was moved to a new
location with more stable geomagnetic conditions. The new SFS observatory is located in Cortijo Garrapilos, Jerez (Cádiz)
and it is operative since 2005. This observatory is a member of INTERMAGNET since 2005 under the IAGA code SFS.
**Ebro Observatory (EBR).** Ebro Observatory was founded in 1904 by the Society of Jesus, with the aim of studying the
Sun-Earth relations. It was located in the town of Roquetes (Tarragona) (Batlló, 2005). The Ebro observatory began to
record periodic measurements of the geomagnetic field in 1905, although the publication of regular results started in 1910
(Observatorio del Ebro, 1910). As noted by Azpiazu and Gil (1919), Ebro Observatory had an excellent location, away from
possible disturbances originated by electric currents, iron masses and geological formations. This observatory had two
pavilions specifically built to carry out geomagnetic measurements. The first one was dedicated to take absolute
measurements with a *Dover* unifilar magnetometer, a *Schulze* dip inductor and a *Plath* galvanometer. The second pavilion
was properly buried and isolated, and it was dedicated to the study of geomagnetic variations. It was equipped with *Mascart*
variometers for the photographic record of magnetic elements.
Ebro Observatory published annual bulletins between 1910 and 1937, when the Spanish Civil war stopped its activity. After
a break of 6 years, it started to work again in 1943, but annual bulletins were not published until 1995. Since 2002, Ebro
Observatory is a member of INTERMAGNET with the IAGA code EBR. Due to electromagnetic interferences produced in
the records because of the city growth, the variometric station was translated in 2001 to Horta de Sant Joan, 20 km away
from the observatory. Since 2012, the measurements are referred to a new main pillar built at Horta de Sant Joan
(Observatorio del Ebro, 2013).

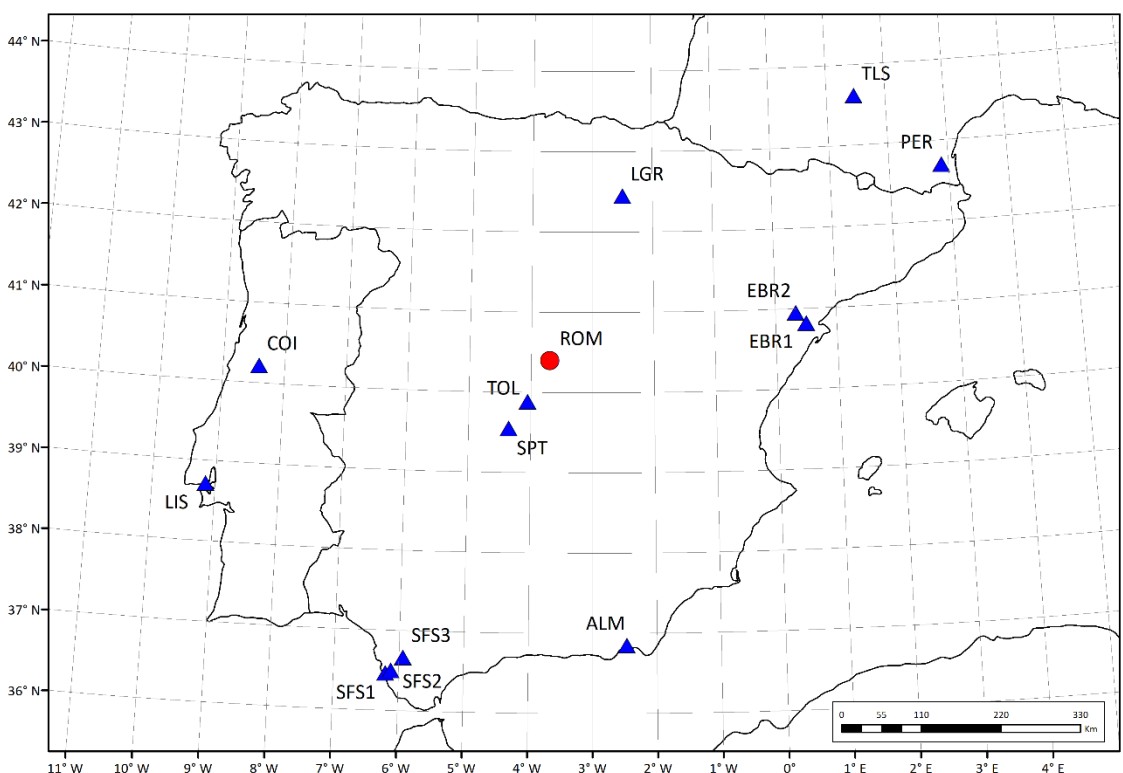

**Figure 7: Location of geomagnetic observatories in the Iberian Peninsula and the south of France.**

**IGN Observatories.** In 1912 the IGN started the works for the generation of the Spanish Geomagnetic Map, that was finally
published for the epoch 1924.0 (Instituto Geográfico y Catastral, 1927). The measurements of the geomagnetic field carried
out along the Iberian Peninsula were referred to Ebro geomagnetic observatory. This observatory was characterized by quite
good quality data but a very eccentric location within the Iberian Peninsula, being located in the northeast corner of Iberian
Peninsula. This circumstance was a problem to consider this observatory as the reference observatory for the national
geomagnetic cartography. Due to this fact, the IGN decided to install its own geomagnetic observatory in the centre of the
Iberian Peninsula. This new geomagnetic observatory was initially projected in the city of Alcalá de Henares, but it was
finally built in the city of Toledo (Azpiazu and Gil, 1919). This marked the beginning of the expansion of geomagnetic
observatories at IGN, a journey that persisted throughout the 20th century.
After the celebration of the International Geophysical Year (IGY, 1st July 1957, to 31st December 1958), the IGN reached
an agreement with the International Union of Geodesy and Geophysics (IUGG) to increase the density of geomagnetic
observatories in Spain. Then, new permanent observatories were stablished in the mainland of Spain, in the cities of Almería
and Logroño. In addition, two more observatories (Miguel Lafuente, 1964) were stablished in Santa Cruz de Tenerife
(Tenerife Island, Canary Islands) and Moca (Fernando Poo Island, Equatorial Guinea), but they are far from the Iberian
Peninsula and have not been taken into account in this work. At present, the IGN has two observatories in operation: one in
San Pablo de los Montes (Toledo) and the other one in Güímar (Tenerife Island, Canary Islands). A brief description of the
mentioned observatories is given below (only for the observatories involved in this study).

**a) Toledo and San Pablo de los Montes observatories.** Taking advantage of the construction of the new Geophysical
Observatory of Toledo in the Buenavista estate on the outskirts of the city, a magnetic section was stablished on it (Sancho
de San Román, 1951; Payo and Gómez-Menor, 1998). In January 1935, the IGN proposed to carry out a new Magnetic Map
of Spain, which was started in 1936. Thus, the works to start the operation of the Toledo Observatory were accelerated to
give assistance to the field measurements (Payo and Gómez-Menor, 1998). The so-called Magnetic Section started to run in
1936 with a set of *Askania* variometers, but the Spanish Civil War produced a cessation of activity since 31st August 1936
up to 1941, when the activity in the geomagnetic observatory was resumed, but providing quite disturbed data due to
conditioning works (Sancho de San Román, 1951). After 1947, the geomagnetic observatory was fully operative, and
yearbooks began to be published without interruption. Besides the *Askania* variometers, the observatory was equipped with a
set of *Topfer* variometers and several instruments to take absolute measurements: one *Schimdt* magnetic theodolite, one
*Askania* terrestrial inductor and one *Carnegie* magnetometer (Payo and Gómez-Menor, 1998). Toledo geomagnetic
observatory was operative until 1981. In the decade of the 1970's, the growth of the city and particularly the railway
electrification, produced significant disturbances over geomagnetic records, mainly in the hours of departure and arrival of
trains to Toledo train station.

For this reason, the IGN projected different magnetic surveys in the Montes de Toledo mountain range to build a new observatory. Finally, a suitable location was found in the town of San Pablo de los Montes, where magnetic anomalies were minimal. In 1974, a plot of 10 Ha was acquired to build the new observatory (Payo and Gómez-Menor, 1998). The construction of this observatory finished in 1978, and a part of the geomagnetic instruments of Toledo Observatory were translated to San Pablo Observatory (SPT according to the IAGA codes). Since then, constant cross-checking work was carried out over a period of two years between both observatories. In 1982, SPT Observatory definitively replaced Toledo Observatory and started publishing their yearbooks. At present, San Pablo Observatory is still in operation and has become the reference observatory of IGN for geomagnetic works. Furthermore, it is a member of INTERMAGNET network since 1992.

**b) Almería Observatory.** In 1949, the IGN decided to expand the Seismic Station of Almería, created in 1911, with a geomagnetic section. New geomagnetic pavilions were projected, whose works ended in 1954 (Morencos, 1964). This observatory was equipped with a set of *La Cour* variometers to record the variations of the geomagnetic field. The absolute instrumentation initially available was one declinometer with an oscillation box by *Sartorius* and one earth inductor by *Wind*. They were updated by a set of *Askania-Werke* instruments: a QHM, a BMZ and an earth inductor. With the new instrumentation, Almería Observatory could take continuous measurements since 1st January 1955. They were published continuously in the yearbooks of the observatory until 1989 when the observatory stopped its activity. The growth of the city of Almería that surrounded the observatory had made that the measurements were highly disturbed.

**c) Logroño Observatory.** Logroño Geophysical Observatory was built by the IGN at 5 km west of this city. The observatory construction started with the geomagnetic pavilion, with the aim of being operative for the IGY. The geomagnetic observatory started to work on 8th July 1957, coinciding almost completely with the beginning of the IGY (Miguel Lafuente, 1964). The instrumentation initially installed at Logroño Observatory was a set of *La Cour* variometers for the record of continuous variations. Besides, there were the following instruments to take absolute measurements: a magnetic theodolite with its oscillation box, a *Sartorius* earth inductor, a torsion magnetometer QHM and a balance magnetometer BMZ. This observatory was continuously running and publishing their yearbooks until 1976, when it stopped its activity.

## 4 Compilation of Declination data

### 4.1 Declination data from Spanish and surrounding observatories

The data from the Spanish observatories described in the previous section have been used in this study to provide declination information at the ROM coordinates. Table 2 summarises information on these observatories, including the period they have been in operation. The yearly mean declination values obtained from the yearbooks published for San Fernando Observatory and Ebro Observatory are shown in Fig. 8a. The monthly mean values of declination of IGN observatories obtained from IGN database are shown in Fig. 8b.


Table 2. Observatories used in this study

| Name | Code* | Country | Latitude (º N) | Longitude (º E) | Altitude (km) | Period | Declination data used** | Distance to ROM (km) |
|---|---|---|---|---|---|---|---|---|
| Real Observatorio de Madrid | ROM | Spain | 40.400 | 356.312 | 0.659 | 1879-1901 | decadal days mean | - |
| San Fernando 1 | SFS1 | Spain | 36.467 | 353.800 | 0.008 | 1880-1979 | yearly mean from WDC | 488 |
| San Fernando 2 | SFS2 | Spain | 36.500 | 353.883 | 0.078 | 1978-2005 | yearly mean from WDC | 482 |
| San Fernando 3 | SFS3 | Spain | 36.667 | 354.067 | 0.06 | 2005-2020 | yearly mean from WDC | 458 |
| Ebro 1 | EBR1 | Spain | 40.817 | 0.500 | 0.532 | 1905-2011 | yearly mean from WDC | 358 |
| Ebro 2 | EBR2 | Spain | 40.950 | 0.333 | 0.532 | 2012-2020 | yearly mean from WDC | 346 |
| Toledo | TOL | Spain | 39.883 | 355.950 | 0.501 | 1947-1981 | monthly mean from IGN database | 65 |
| San Pablo de los Montes | SPT | Spain | 39.550 | 355.650 | 0.917 | 1982-2020 | monthly mean from IGN database | 110 |
| Almería | ALM | Spain | 36.850 | 357.533 | 0.065 | 1955-1989 | monthly mean from IGN database | 408 |
| Logroño | LGR | Spain | 42.450 | 357.500 | 0.445 | 1957-1976 | monthly mean from IGN database | 249 |
| Lisbon | LIS | Portugal | 38.717 | 350.850 | 0.1 | 1858-1900 | yearly mean from WDC | 504 |
| Coimbra | COI | Portugal | 40.217 | 351.583 | 0.099 | 1867-2020 | yearly mean from WDC | 401 |
| Toulousse | TLS | France | 43.617 | 1.467 | 0.154 | 1882-1905 | yearly mean from WDC | 557 |
| Perpignan | PER | France | 42.700 | 2.883 | 0.037 | 1886-1910 | yearly mean from WDC | 606 |

*All codes are IAGA codes except for the ROM code
** WDC = World Data Centre; IGN = Instituto Geográfico Nacional

In our study, we have also considered the declination measurements made at other geomagnetic observatories near Spain,
situated in Portugal and southern France. In Portugal, the geomagnetic observatory with greatest tradition recording and
measuring the Earth's magnetic field is the one of the *Instituto Geofísico da Universidade de Coimbra*. This observatory
started to work in 1866, although in 1931 it had to be translated to a new location in Alto de Balaia Street to avoid the
disturbances induced by the power lines (Custodio de Morais, 1953). This observatory is still working today (as COI in the
IAGA codes), so it has the longest geomagnetic measurements series of the Iberian Peninsula and one of the longest series in
the world. The annual mean values of this series are published in the World Data Centre for Geomagnetism (WDC) and are
continuously updated. A homogenised revision of the Coimbra observatory data (Morozova, 2021) has recently been
published, but not significant differences are observed for the purpose of our study, and thus, we have considered the
previous data published by the WDC. The declination values of this series are shown in Fig. 8c. Besides, geomagnetic
measurements were made in Portugal, in the city of Lisbon, since the year 1858, at *Observatorio do Infante D. Luiz*
(Observatorio do Infante D. Luiz, 1863). This observatory published since that year the annual results of its measurements of
the different components of the geomagnetic field, and it was operational until 1900. The installation of electric lines for the
tram near the observatory disturbed the normal operation of the magnetic instruments and it was impossible to use their
measurements since that date (Observatorio do Infante D. Luiz, 1904). The declination values of this series, extracted from
the yearbooks published by this observatory, are shown in Fig. 8c. Information of these observatories in Portugal is shown in
Table 2.
In the south of France, there were also two geomagnetic observatories located in the cities of Toulouse and Perpignan. They
started to record the geomagnetic field at the end of the 19th century, but they stopped at the beginning of the 20th century,
so the measurement series of them are very short. The annual mean values of the geomagnetic components measured at these
observatories are also available at the WDC. For Toulouse Observatory, the series begin in 1882, although it only has
continuity between 1894 and 1905. For Perpignan Observatory, the series cover the period from 1886 to 1910, although it
presents a gap of data between 1902 and 1906 (see Table 2). Figure 8c also shows the declination values corresponding to
these observatories in southern France.

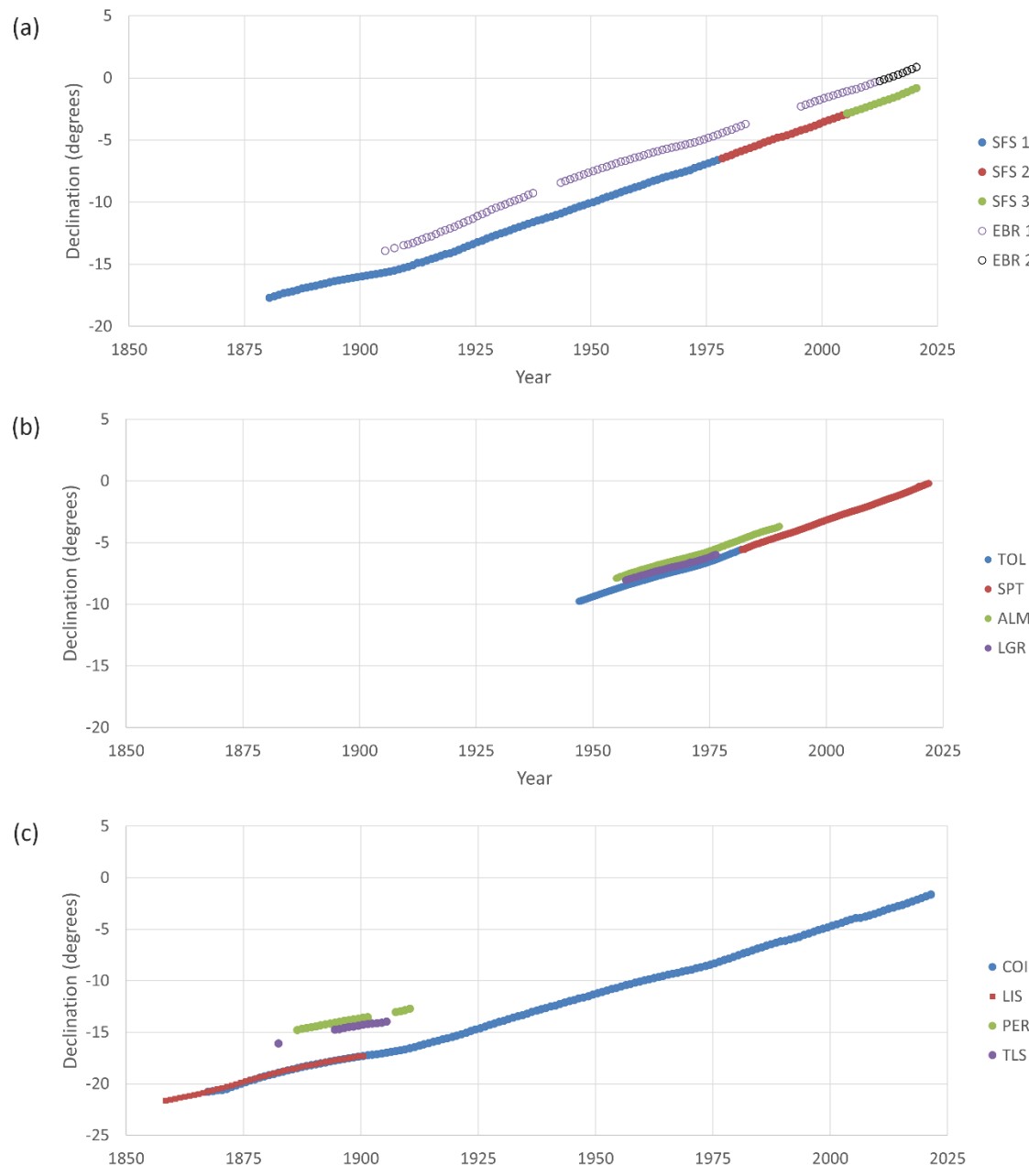


**Figure 8: Evolution of magnetic declination: (a) San Fernando and Ebro observatories, (b) IGN observatories, (c) surrounding observatories.**

## 4.2 Declination historical data

Based on the previous compilation, the recorded geomagnetic observatory data in the Iberian Peninsula and surrounding areas date back to the latter half of the 19th century. These records offer a good temporal coverage, spanning from that period to the present day. In order to add more information of declination data prior to the observatories epoch, we have considered the information available at the HISTMAG database (Arneitz et al, 2017). This database has integrated a large amount of historic geomagnetic data from all around the world, including archaeomagnetic and volcanic data. The historical compilation is mainly based on the previous compilation of Jonkers et al. (2003) that bring together a huge amount of data obtained at naval trips with measurements made on land. In addition, HISTMAG completed the Jonkers' database with historical information from other sources that include measurements made for mining, sundials, cartography, etc. For the purpose of this work, we have made a query on HISTMAG database, considering a spherical cap with centre at ROM and radius of 1000 km. The historical declination data covers the period from 1500 to 1900. Figure 9 shows a map with the spatial distribution of the selected data.

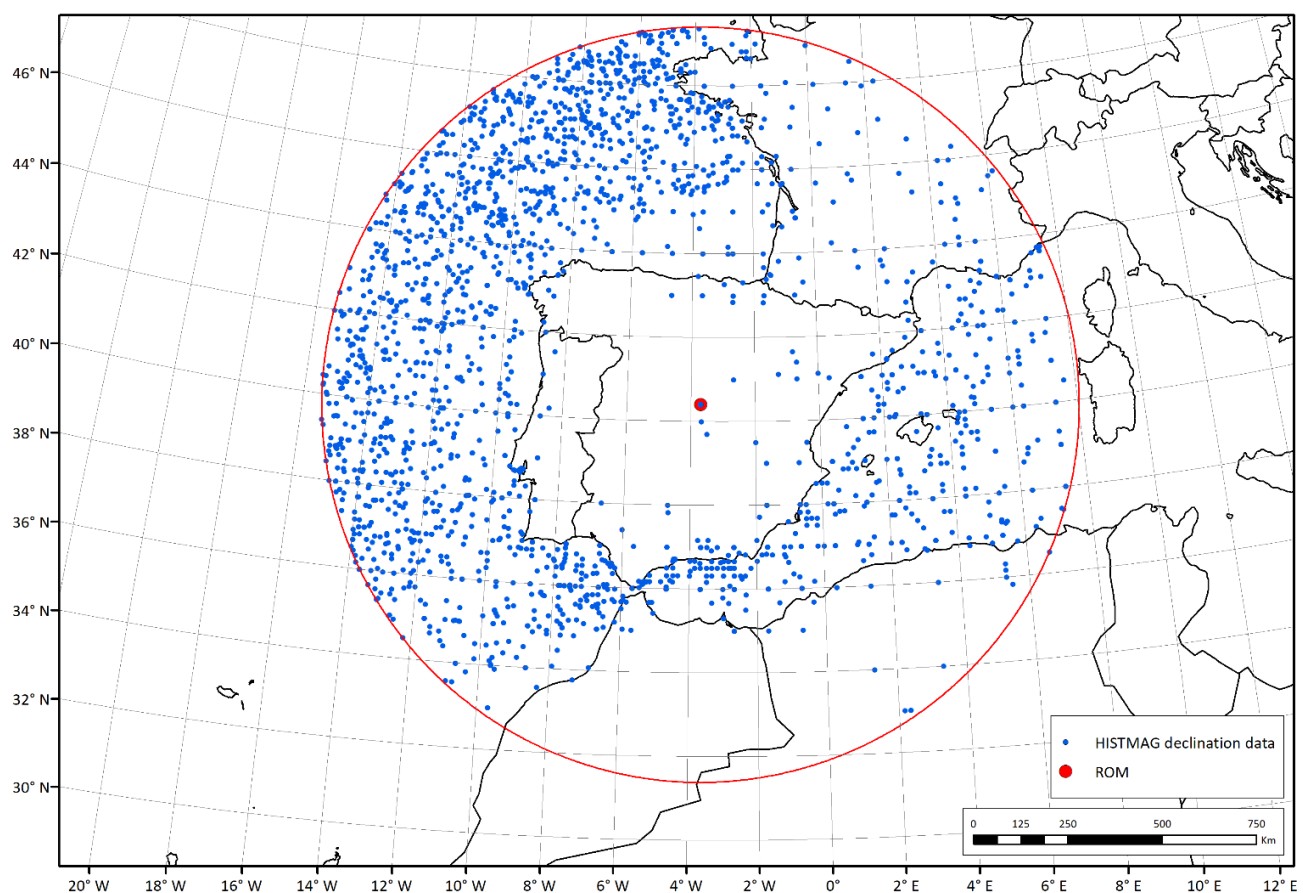

**Figure 9: Declination points selected from HISTMAG database. Red point corresponds to the ROM coordinates (40.4000º N, 3.6879º W). The radius of the spherical cap is 1000 km.**


The result of the query provided a total of 3512 declination records. To check the initial quality of these data, we have made
a comparison of them with the data provided by the geomagnetic model *gufm1* using the coordinates of the points and the
dates of their records, extracted from the database. The results, in terms of declination residuals, are shown in Fig. 10. The
residuals follow a normal distribution, centre in 0.05° and standard deviation of 1.68°. This was expected since the major part
of these data were used in the construction of the *gufm1* model. Therefore, we have considered that these data are suitable to
be used in this study.

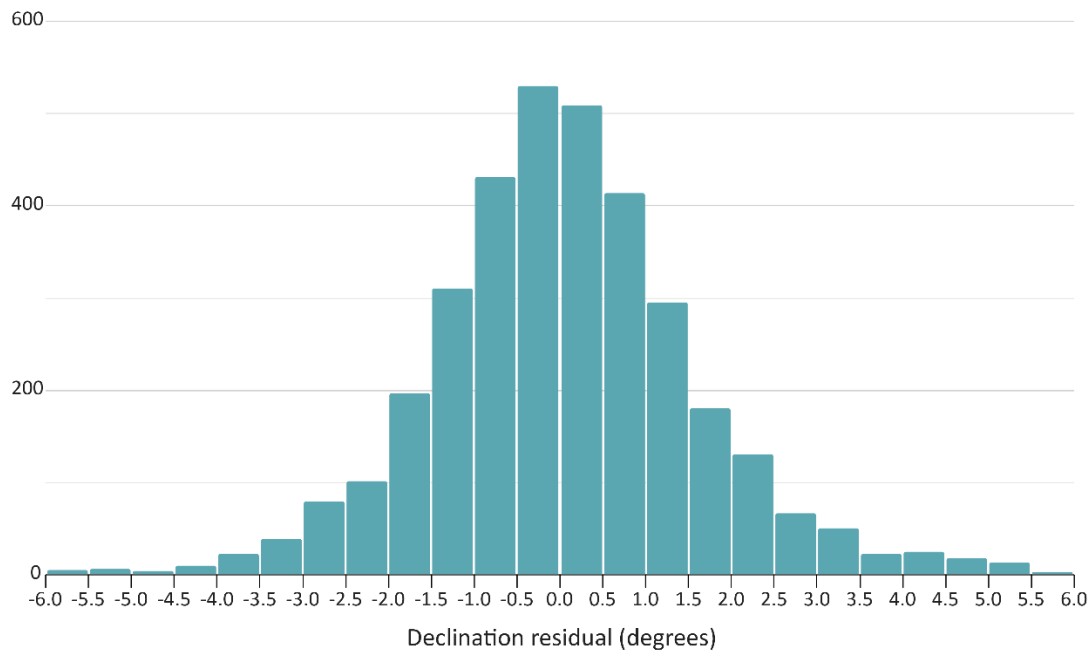


**Figure 10: Residuals from the comparison between HISTMAG declination data with those given (at the same time and location) by**
**the *gufm1* model.**
**4.3 Transfer of all the declination data to ROM coordinates**
As indicated in the previous sections, the magnetic declination measurements recorded in the ROM are very scarce. They
only cover some decades at the end of the 19th century. For this reason, if one wants to analyse the time evolution of the
declination element at the ROM coordinates, we need to transfer the rest of the declination measurements (i.e., the
observatory data from the Iberian Peninsula and the south of France, and all the historical data of the HISTMAG database)
from the original locations to the ROM coordinates. This declination database will provide information about the declination
at the ROM coordinates over the last 450 years. To transfer the declination data from the original locations to the ROM
coordinates (40.4000° N, 3.6879° W) we use the declination spatial gradient estimated from the *gufm1* model from 1590 to
1840 and the most recent model Cov-Obs.x2 (Huder et al., 2020) from 1840 to the present days. To do that, we estimate for a
certain time the difference in declination for the original location and the value given at the ROM coordinates. Then this
difference, taken as a spatial gradient, is added to the original declination data, providing the transferred value. In Fig. 11, we
show the value of the declination gradient within the spherical cap of Fig. 8 for four different epochs (1600, 1750, 1900, and
2020) according to these two geomagnetic models.

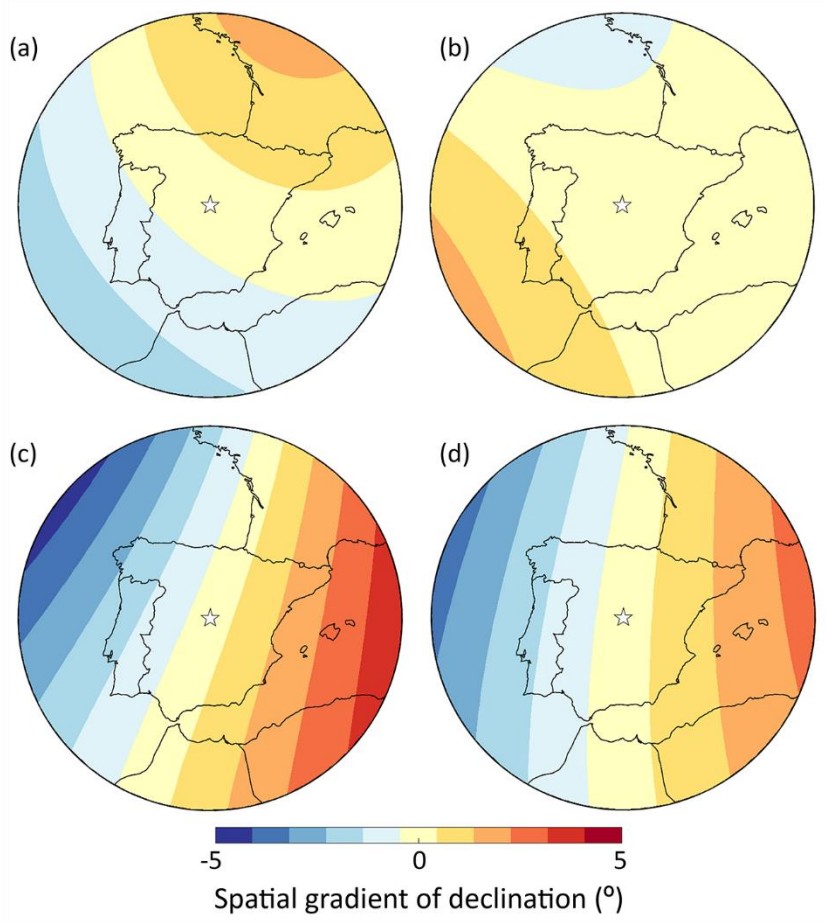


**316 Figure 11: Spatial gradient map of the declination at four different epochs. Maps at (a) 1600 and (b) 1750 were estimated by *gufm1***
**317 model and (c) 1900 and (d) 2020 maps by the Cov-Obs.x2 model. The white star corresponds to the ROM coordinates (40.4000º N,**
**318 3.6879º W).**

Figure 12a shows the result of applying the spatial gradient method to the historical data obtained from the HISTMAG
database. In Fig, 12b, the transferred declination data from the IGN observatories (Toledo, Almería, Logroño and San Pablo
de los Montes) from which we have monthly mean declination values are shown. In addition, Fig. 12c shows the same result
for data measured at other observatories of the Iberian Peninsula and south of France (San Fernando, Ebro, Coimbra, Lisbon,
Perpignan and Toulouse) from which the annual mean declination values are available. The transferred declination data
reveal a clear difference between the observatory data and the historical observations compiled in HISTMAG. The historical
observations transferred to ROM exhibit significant dispersion (Fig. 12a) due to their inherent characteristics (see Jackson et
al., 2000). However, the observatory data show good agreement after being relocated to ROM coordinates (Fig. 12b, c).

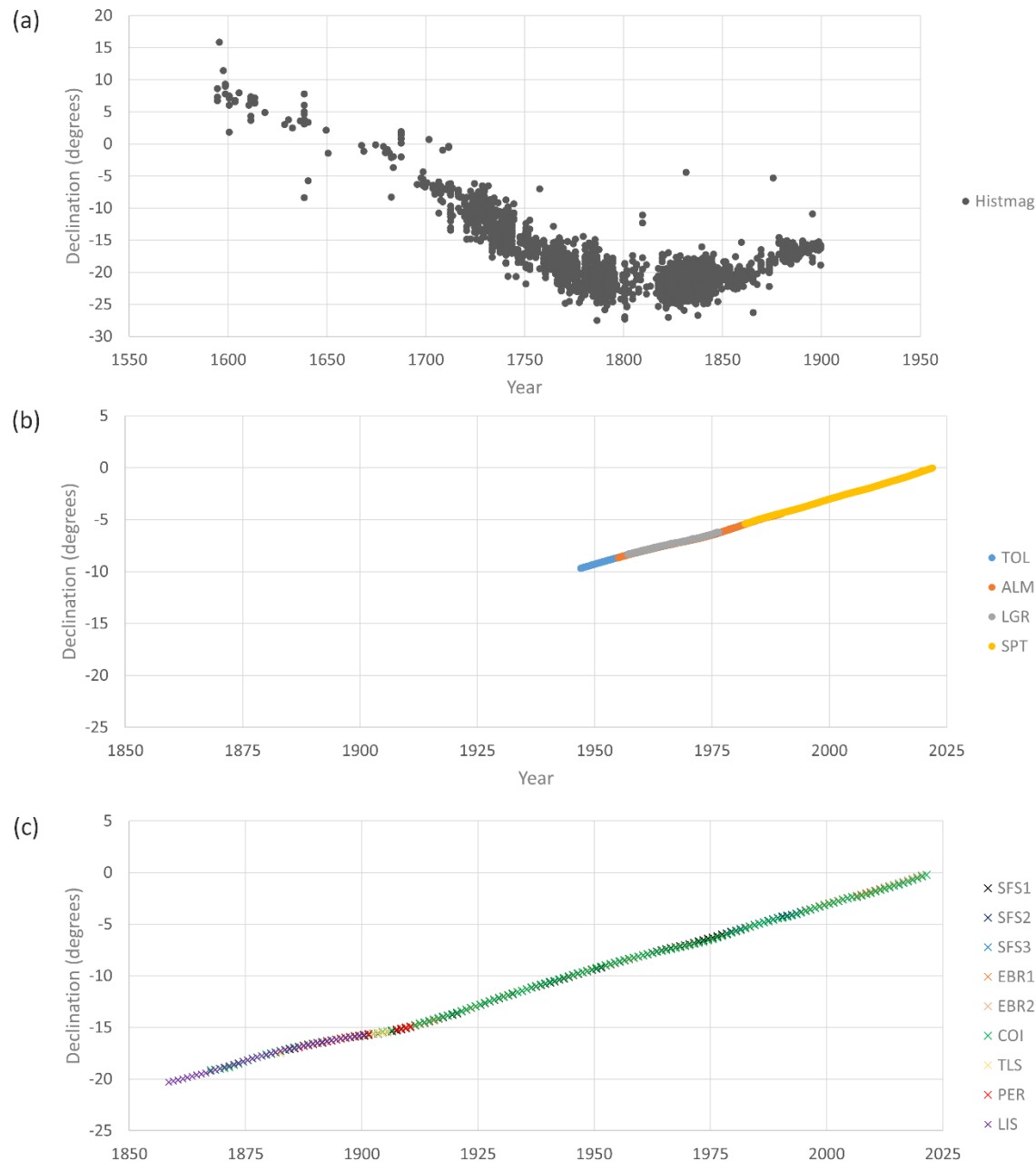


**Figure 12: Declination data transferred to the ROM coordinates, which have been separated into different panels for a better visualization: (a) HISTMAG historical data, (b) IGN Observatory data, (c) Other Observatory data.**

## 5 Results and discussion

### 5.1 Declination curve at ROM

With the declination data transferred to the ROM coordinates shown in Fig. 12, we have generated a time-continuous curve
for the declination from 1590 up to the present days. To obtain the curve, we have applied a bootstrapping method (similar to
that of Thébault and Gallet, 2010) taking into account the declination uncertainty of each individual datum. In the curve
construction, the temporal domain is expressed by means of cubic B-splines with knot points every 5 yr. The set of data have
been ranked into two categories: the historical data that covers from the earliest times up to 1900 and the instrumental series
covering from 1900 to the most modern values. To provide a smooth declination curve, the cubic B-splines are penalized by
minimizing the second time derivative of the declination curve by means of a damping temporal parameter. The optimal
value obtained for the historical data was $\lambda = 0.1$, and for the instrumental series was $\lambda = 0.001$.
To get the error bars of the declination curve, the bootstrap approach considers 1000 sets of data generated bootstrapping the
data considering their measurement uncertainties. In this sense, we have considered three different declination uncertainties
according to the following periods. In the interval since 1900 to the present date we have considered an uncertainty of 1
minute of arc taking into consideration the accuracy of the declinometers used in the Spanish observatories during the 20th
century (Batlló, 2005) and the analysis of the hourly mean values uncertainty carried out by Curto (2019). It is difficult to
properly know an uncertainty value for the declination values before 1900. In relation with the data of historical values of
declination collected at the HISTMAG database, we do not know the uncertainty of the compasses used in the measurement
of the declination. According to Jackson et al. (2000), who include a noise error of 0.5º for these historical observations, we
have used this value as uncertainty for the declination data before 1900. Although some of these data belong to the earliest
observatories functioning in the Iberian Peninsula, no detailed information is available about the uncertainty of their
measurements, so we have decided to be conservative and use the same value. Being even more conservative, we have
decided to double this uncertainty value (i.e. 1º) for the historical declination data prior to 1750, so the accuracy of the
measurement and the resolution of the compasses are supposed to be lower as we go back in time.
For each bootstrapped dataset, we generate a declination curve. The final curve is the mean of the 1000 obtained curves and
the error bands (at 1σ of probability) are obtained using the standard deviation of the 1000 curves. As result, we get the
declination curve for the ROM plotted in Fig. 13.

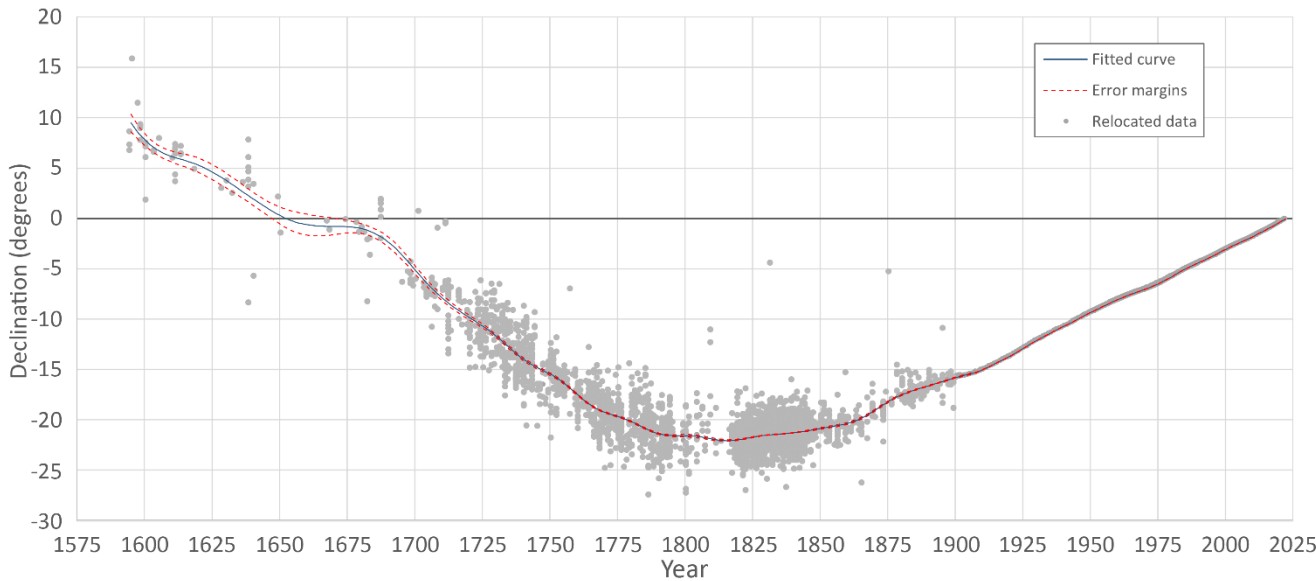


**Figure 13. Fitted declination curve obtained for the Royal Observatory of Madrid and its error margins at 1σ. At the background, all transferred (or relocated) historical and instrumental data used for curve fitting are plotted by grey dots.**

This fitted curve shows that at the Royal Observatory of Madrid the minimum declination value achieved in the period of study was -21.99° in the year 1816. Since then, the value of declination at that location has been continuously increasing until reach positive values at the beginning of the year 2022. Before the minimum, the declination value had been decreasing since the beginning of the selected period (year 1590) and the previous crossing of the agonic line would have taken place around 1652 changing declination from positive to negative values, being 1647-1671 the period of 95% probability (see Fig. 14).

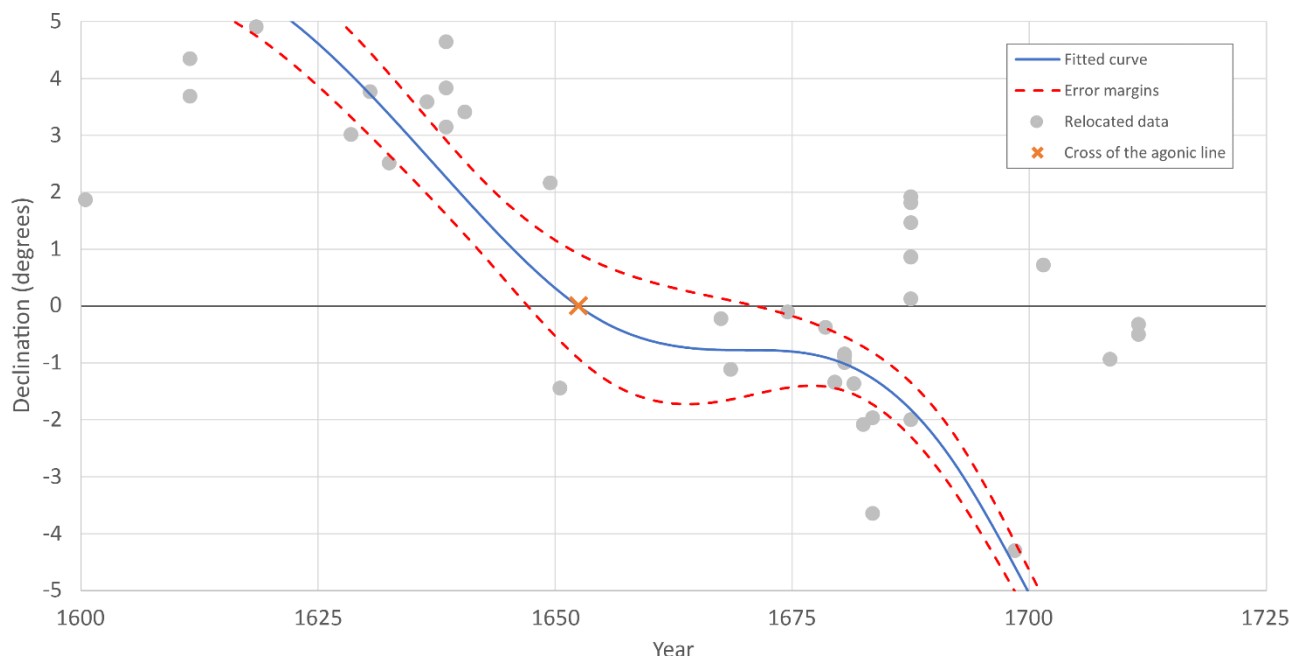

366

**Figure 14: Detail of the declination curve obtained showing the crossing of the agonic line by the Royal Observatory of Madrid around the year 1652. Red dashes lines show the error margins of the declination curve. At the background, transferred (or relocated) data that have been used for curve fitting.**

With the optimal declination curve obtained for the ROM, the residues of each group of data (i.e., historical and instrumental data) used in the calculation process with respect to the fitted curve have been calculated (Fig. 15). For the historical data, the histogram of residual data points out the contribution of two type of distributions: a Gaussian distribution plus a Laplacian distribution, both centred at 0° (Fig. 15a). For the instrumental data, the histogram follows a Gaussian distribution centred at 0° (Fig. 15b). These results indicate an appropriate fitting of both series of data to obtain the declination curve at the ROM coordinates. As expected, the high dispersion of the historical data (see, e.g., Fig. 13a) is evident in the greater width of the residual data distribution compared to that of the instrumental data series.

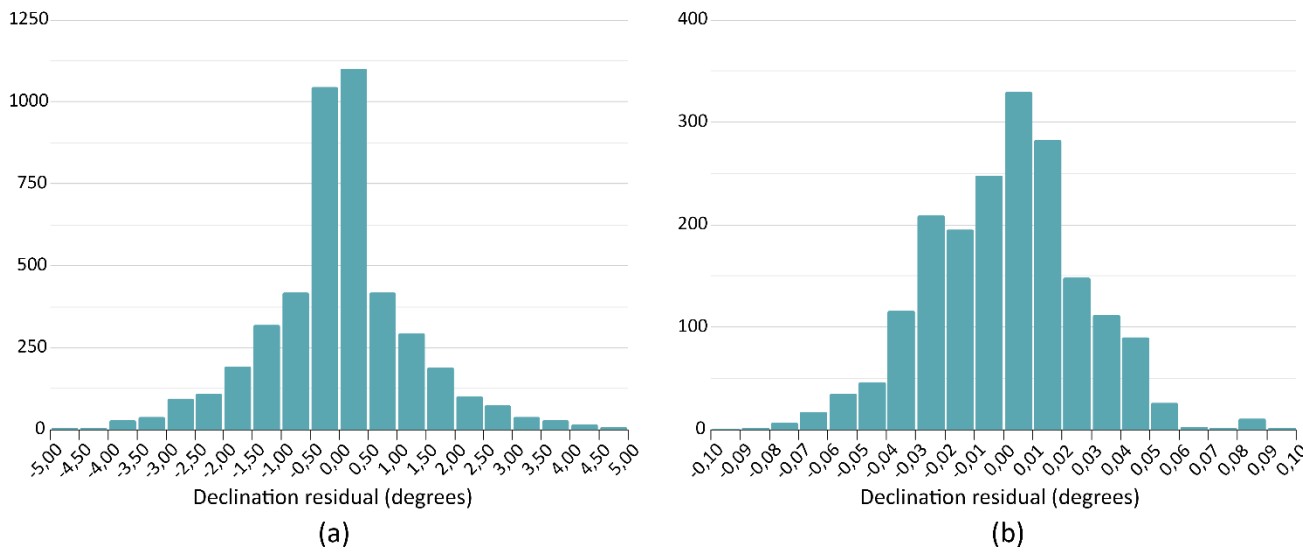

(a)                      (b)

**Figure 15: Distribution of residuals between original data and the fitted declination curve of the ROM: (a) Histogram of residuals for the historical data series; (b) Histogram of residuals for the instrumental data series.**

The secular variation curve (i.e., the first time derivative) for declination has been calculated from the ROM declination curve previously obtained. This curve (Fig. 16a) illustrates the non-constant nature of the secular variation in declination over time, displaying significant temporal variability initially related to processes in the deep Earth's interior, where the geomagnetic field is originated. However, a detailed analysis of this variability reveals a clear periodicity characteristic of external solar forcing, specifically the solar cycle. The secular variation declination curve demonstrates a pronounced influence of the external geomagnetic field, modulated by the 11-year and 22-year solar cycles. This external influence has not been adequately removed from the original declination data. To mitigate the effect of solar activity, we applied a filter that removes periods shorter than 25 years, and the filtered curve is shown in Fig. 16a. The contribution of solar activity is depicted in Fig. 16b. It is important to note that solar activity is not accurately recorded before 1700 due to the limited number of declination data points (see Fig. 13). This finding highlights the necessity of filtering geomagnetic observatory data to eliminate any residual contributions from the external field when analysing long-term time series.

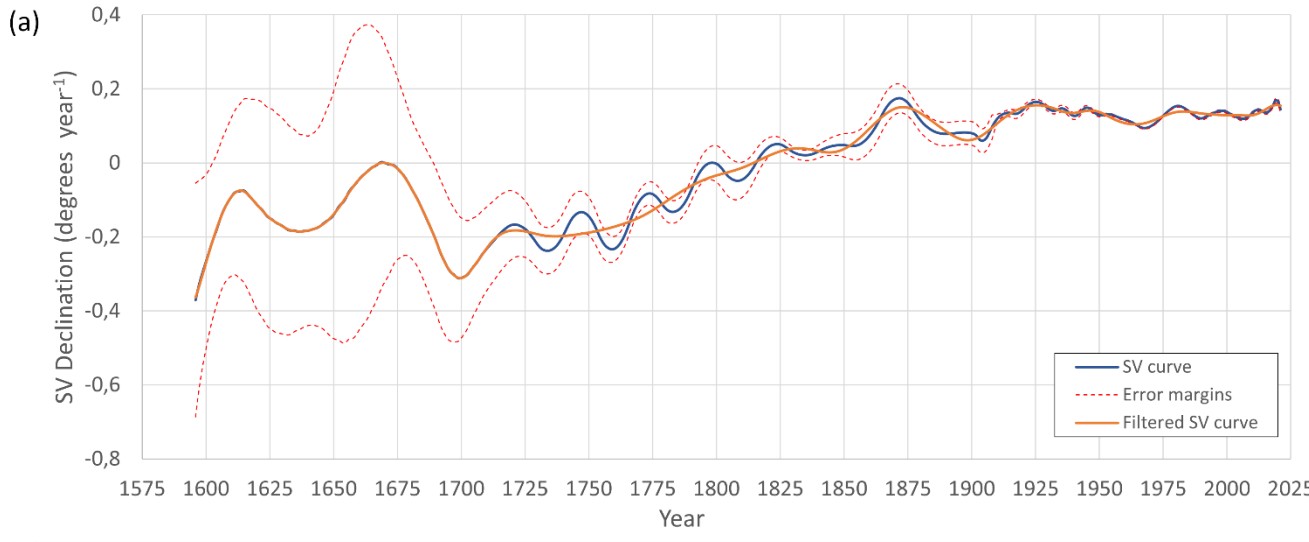

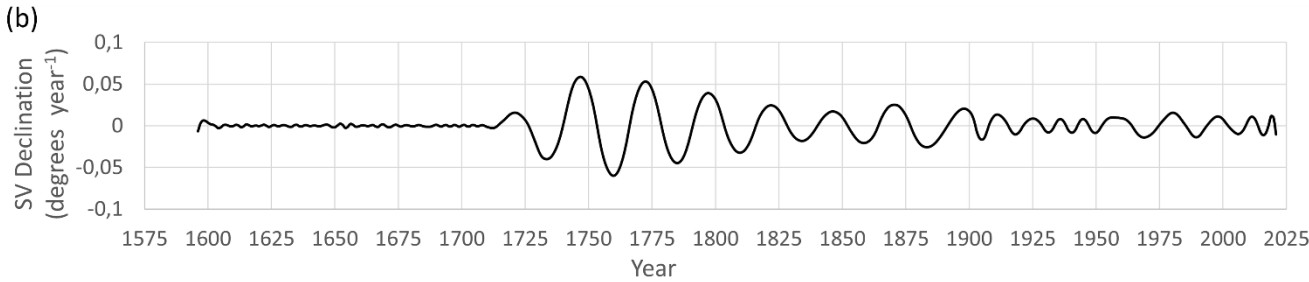

Figure 16. (a) Secular variation curve obtained for the Royal Observatory of Madrid and its error margins at 1σ of probability and filtered secular variation curve removing periods shorter than 25y (b) Solar contribution to the secular variation curve (residual between the blue and orange curves in (a)).

## 5.2 Comparison with independent data and historical global models

In order to check the validity of the obtained declination curve, we have compared the curve with a compilation of declination data not included in the curve that cover a period before 1855. These independent declination data correspond to the measurements made by Rico Sinobas (1856) at different locations within the Iberian Peninsula. From this compilation (see S4 in Supplementary Material), we have selected the observations from 1600 onwards and discarded two observations whose location is badly defined. The coordinates of these selected points have been determined and the observations have been transferred to the ROM coordinates. The transferred declination data are listed in Table 3.

Table 3. Declination values compiled by Rico Sinobas and their value transferred to the ROM coordinates.

| Location | Date | Latitude (°N) | Longitude (°W) | Declination (°) | Declination transferred to ROM (°) |
|---|---|---|---|---|---|
| Lisboa | 1638.5 | 38.7080 | 9.1390 | 7.65 | 7.81 |
| Lisboa | 1668.5 | 38.7080 | 9.1390 | -0.83 | -1.12 |
| Valencia | 1675.0 | 39.4700 | 0.3764 | 0.00 | 0.35 |
| Lisboa | 1683.5 | 38.7080 | 9.1390 | -30.00 * | -30.64 |
| Lisboa | 1697.5 | 38.7080 | 9.1390 | -4.30 | -5.27 |
| Lisboa | 1706.5 | 38.7080 | 9.1390 | -6.50 | -7.59 |
| Valencia | 1707.5 | 39.4700 | 0.3764 | -5.00 | -4.42 |
| Cádiz | 1724.5 | 36.5350 | 6.2975 | -5.42 | -6.16 |
| Gibraltar | 1733.5 | 36.1400 | 5.3500 | -13.63 | -14.17 |
| Cabo de Gata | 1733.5 | 36.7219 | 2.1930 | -13.93 | -13.93 |
| Cabo de San Vicente | 1733.5 | 37.0250 | 8.9944 | -13.82 | -14.96 |
| Cabo de Santa María | 1734.5 | 36.9602 | 7.8871 | -14.33 | -15.26 |
| Madrid | 1747.5 | 40.4000 | 3.6879 | -16.50 | -16.50 |
| Gibraltar | 1761.5 | 36.1400 | 5.3500 | -17.18 | -17.68 |
| Lisboa | 1762.5 | 38.7080 | 9.1390 | -17.53 | -18.00 |
| Cádiz | 1769.5 | 36.5350 | 6.2975 | -17.25 | -17.79 |
| Cádiz | 1771.5 | 36.5350 | 6.2975 | -18.00 | -18.55 |
| Cádiz | 1776.5 | 36.5350 | 6.2975 | -19.70 | -20.27 |
| Lisboa | 1776.5 | 38.7080 | 9.1390 | -19.00 | -19.22 |
| Lisboa | 1782.5 | 38.7080 | 9.1390 | -19.85 | -19.99 |
| Madrid | 1785.5 | 40.4000 | 3.6879 | -20.00 | -20.00 |
| Cádiz | 1791.5 | 36.5350 | 6.2975 | -21.93 | -22.51 |
| Madrid | 1799.5 | 40.4000 | 3.6879 | -22.03 | -22.03 |
| Madrid | 1804.5 | 40.4000 | 3.6879 | -21.50 | -21.50 |
| Cádiz | 1807.5 | 36.5350 | 6.2975 | -22.50 | -23.00 |
| Lisboa | 1811.5 | 38.7080 | 9.1390 | -22.75 | -22.30 |
| Lisboa | 1820.5 | 38.7080 | 9.1390 | -22.70 | -22.18 |
| Lisboa | 1829.5 | 38.7080 | 9.1390 | -22.38 | -21.81 |
| Lisboa | 1853.5 | 38.7080 | 9.1390 | -22.38 | -21.18 |
| Cartagena | 1853.5 | 37.6000 | 0.9819 | -18.88 | -20.37 |
| Málaga | 1853.5 | 36.7167 | 4.4167 | -20.18 | -20.79 |
| Cádiz | 1853.5 | 36.5350 | 6.2975 | -21.93 | -22.05 |
| Santander | 1853.5 | 43.4667 | 3.8000 | -21.22 | -20.34 |

* This value seems to be a misprint. For a most probable value of 30' W the reduced declination
would have a value of -1.14°.

The result of this comparison is shown in Fig. 17a, where it can be seen that they fit quite well with the declination curve
obtained for ROM. Only one observation corresponding to Lisbon for the year 1683 shows a great discrepancy with the
declination curve. It seems to be a typo in the original document as the measurement of a declination value of 30º00' W in
the Iberian Peninsula has not been reached in the whole period studied. It appears that a value of 30' W might have been
more likely and would align with the declination curve, as illustrated in Fig. 17a by a red triangle. This comparison indicates
that the compilation made by Rico Sinobas has enough quality and it could be taken into account for future declination
studies.
We have also compared the declination curve obtained for the Royal Observatory of Madrid with the declination values
given by the geomagnetic *gufm1* and *Cov-Obs.x2* models for this location. In Fig. 17b, the ROM declination curve has been
plotted together with those provided by the *gufm1* model for the period 1590-1840 and the *Cov-Obs.x2* model for the period
1840-2022. The obtained declination curve and those synthetized by both models show a good agreement, specially after
1800 where the amount of declination data increase. Before this epoch, the discrepancy increases due to the small amount of
data measured in that time and the observed high dispersion.
In terms of secular variation, we compared the original curve and the filtered curve with the model predictions (Fig. 17c).
Observing the original secular variation curve (blue line in Fig. 17c), we note a clear agreement with the secular variations of
the global models. The largest discrepancies occur before 1800, a period where the curve and the *gufm1* model are
constrained by limited declination data. The comparison with the filtered curve (orange line in Fig. 17c) highlights the
impact of solar forcing on the global models. Both the *gufm1* and *Cov-Obs.x2* models replicate the time variability of the
original curve. These results could indicate that the historical models do not adequately account for the influence of solar
forcing in their construction.

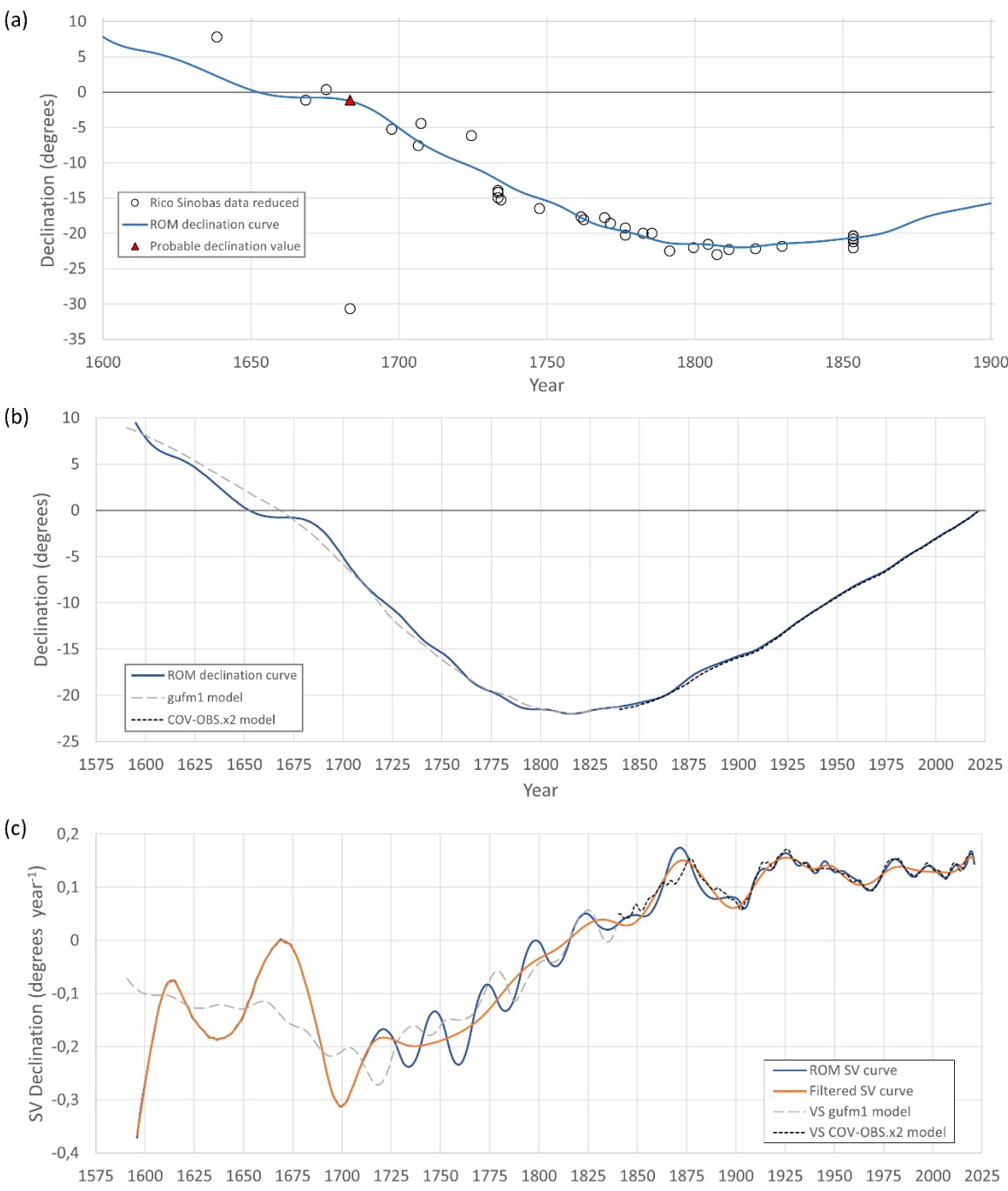


Figure 17. (a) Declination curve obtained for the Royal Observatory of Madrid and declination values collected by Rico Sinobas
and reduced to the ROM coordinates. The probable value for the wrong declination of Lisbon in 1683 is shown by a red triangle.
(b) Declination curve obtained compared with the declination predicted by gufm1 and Cov-Obs.x2 models. (c) Obtained and
filtered secular variation curves compared with the secular variation predicted by gufm1 and Cov-Obs.x2 models.

## 6 Conclusion

The Royal Observatory of Madrid was established by King Carlos III in 1785, with construction beginning around 1790. However, it did not become operational until 1851 due to various challenges. In 1853, the Observatory expanded to include meteorological and geomagnetic observations, acquiring several specialized instruments. In September 1855, Mr. Rico Sinobas made the first continuous geomagnetic measurements in the Iberian Peninsula. Discrepancies in early geomagnetic data were noted, possibly due to metallic influences from the Observatory's construction.

In December 2020, the agonic line crossed the ROM. This event has prompted this work with a comprehensive study of the declination behaviour at ROM coordinates over the past four centuries. To achieve this, we processed declination data from the Iberian Peninsula and nearby regions, collected from geomagnetic observatories since the late 19th century and older historical data compiled in the HISTMAG database. This allowed us to create a declination curve for the Royal Observatory of Madrid, pointing out how the agonic line also crossed the ROM around 1652. The obtained curve aligns with independent declination data measured by Rico Sinobas in the Iberian Peninsula during the last century. Additionally, our results highlight the significant influence of solar forcing on the declination curve, reflecting the impact of the solar cycle on the secular variation of the declination. This effect has also been observed in other historical global models, where this external forcing has not been adequately mitigated.

## Author contribution

Conceptualization, J.M.T. and F.J.P.-C.; data curation, J.M.T. and F.J.P.-C; formal analysis, J.M.T. and F.J.P.-C.; investigation, J.M.T.; methodology, F.J.P.-C.; software, F.J.P.-C.; supervision, F.J.P.-C. and A.B.A; validation, J.M.T. and F.J.P.-C.; visualization, J.M.T, A.N. and F.J.P.-C; writing – original draft preparation, J.M.T.; writing – review and editing, F.J.P.-C., A.N. and A.B.A. All authors have read and agreed to the published version of the manuscript.

## Competing interests

The authors declare that they have no conflict of interests.

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
