# Peer review of "Historical evolution of the geomagnetic declination at the Royal Observatory of Madrid."

_History of Geo- and Space Sciences, 2024_

## Author Comment (AC4)

Dear editor *Dr. Kristian Schlegel.*

Thank you very much for your recommendations. We have followed the Referees' suggestions, including the revision of the issues you highlight. We hope that the manuscript is now clearer to the reader. Below, we provide the point-by-point answers to the comments and suggestions (comments are in black italic letters and our answers in blue letters).

**Response to Reviewer 1:**

We would like to thank *Dr. Mioara Mandea* for her comments, which have helped to improve the quality of the manuscript.

As a result of the review, we have reorganized some sections of the manuscript giving more importance to the role of the ROM in this work. We have included new tables with useful information about ROM and new photos of the geomagnetic instrumentation preserved in the ROM. We have also included a new figure (new Figure 7) and made some corrections to other figures for optimal display. Other corrections on the text and results have been made to improve the manuscript.

Bellow we include a detailed explanation, point-by-point, to each of the specific comment or suggestion raised by the reviewer.

**Specific comments**

*- The paper structure needs to be revisited, with a clear introduction, a section dedicated to ROB where all information about it are gathered, a section on other Spanish observatories, a result session to show different comparisons between observatories and with models, before concluding.*

We agree with your comment. We have proceeded to restructure the manuscript according to your indications. Sections about observatories have been reorganized highlighting the importance of the ROM. All the observatory data are now compared in the same section and the comparison with global models is carried out in the discussion section. We hope that the new structure will be clearer and easier to understand.

*- This manuscript invokes to provide an historical evolution of the geomagnetic declination at the Royal Observatory of Madrid. In describing the history it is needed to tell more about the first instruments (e.g. lines 74-77 – which kind of magnetometers of inclinometers, etc). Figure 5 shows an inclinometer, maybe some others photos can be provided?*

We have tried to include more information about the magnetic instruments used at ROM, but very little documentation is available. We have now included a table summarising the instrumentations at ROM and some new references about them. New pictures with the original instrumentations at ROM have been also included.

*- Many information are difficult to read. For example, in the section "Observatory data selection" I suggest the authors to draw a map with the used observatories. It would be nice to provide a table gathering the main information (observatory name, iaga code, coordinates, starting time - closing time, instruments, data availability - minutes, hourly, monthly, annual, etc).*

To facilitate reading, we now include a map with the location of the observatories and a table with more information about the observatories and the data that we have used to develop the declination curve. However, we have not included detailed information on the instruments used in the other observatories during this period, because this is not the aim of our work, which focuses on the history of the Royal Observatory of Madrid.

*Some figures need to be revisited. E.g. In Figure 4 the comparison with the gufm1 model is presented, but this model is not built to describe variations on daily basis. Moreover, the difference is not du only to crustal field, but maybe to other factors, as unmodelled contributions or errors in measurements. To compute the crustal field (which can be considered constant in time) I suggest you to adapt methods proposed in the literature.*

The referee is right. The *gufm1* model provides long time-scale information (annual VS) and does not provide information on the daily changes that are present in the ROM data in this figure. We have now modified the figure, showing only the measured declination data. In the text, we indicate that there is a difference of about 2º with the value predicted by the *gufm1* model for that epoch.

*- Figures indicating series and comporting many plots may be view over the same time-span (min _ max of the series). This refers mainly to figures 7, 11.*

We have followed your instructions and homogenised the axis limits in the different multi-panels figures for a better comparison between them.

*- Coimbra observatory has a long series of observation. A comparison with the series published in 2021 (and not cited), can bring interesting information and discussions.*

Thank you for drawing our attention to the work of Morozova et al. (2021). We have included in the manuscript a reference to this publication that contains a homogenised dataset of the Coimbra observatory. However, we did not use the homogenized declination series in our process since no significant changes were observed for our purposes. As shown in the next figure, both the original series from the WDC and the homogenized series closely agree with each other. We found a dispersion ranging from 0º to 0.1º, which does not provide additional information relevant to our work.

[Figure]

**Figure 1.** Declination data from COI: data from WDC (blue stars) and the homogenised time series (red circles) of Morozova et al. (2021).

*- The IGRF model is invoked (line 369), without citations. This choice is not very clear, and I wonder why not to use the gufm1 and COV-OBS models, only.*

Thank you for your advice. We have now rectify that mistake. In an earlier version of the paper, we used the *IGRF-13* model for our comparisons, but then decided that using the *COV-OBS.x2* model was more convenient. Therefore, we have revised and homogenised the references to the *gufm1* and *COV-OBS.x2* models in the text and figures.

*- The last session needs to be revisited, to get into some real discussions/ conclusions. The section on ROM observatories has to include the Table 1 and the Figure 16. At the end of the paper we would expected some interpretation of the observed secular variation curves.*

As suggested by the referee, the last section has been rewritten. The section with the main results now includes two subsections: one about the declination curve (and its secular variation) obtained at ROM, and other comparing it with independent data and global models. In both cases, we have included more information about the curves and the comparison with data and models.

*Finally, efforts are needed to improve the English. Some parts are difficult to read: e.g. abbreviation used and not defined or defined after some pages, as IGN; "Declination data is translated from SPT to ROM » - data are not translated, they are adjusted; some expressions are not clear ("Note that the solar activity is not recorded before 1700 due to the scarce number of declination data"). I encourage the authors to work on the manuscript to provide a more fluid reading.*

We have revised and improved the English in the manuscript to make it more readable. We addressed the expressions you indicated and defined all abbreviations upon their first appearance. We hope that the text is now easier for readers to follow.

**Response to Reviewer 2:**

We would like to thank *Dr. Tohru Araki* for his comments, which have helped to improve the quality of the manuscript. We have considered your specific comments and modified the figures affected and the text of the manuscript.

Bellow we respond to each of the comments raised by the reviewer.

**Specific comments**

1. *The font on the diagrams is too small.*

   Thank you for the warning. We have revised all figures, using larger fonts to improve readability.

2. The full names should be included in parentheses when their abbreviations appear at first.

   We have revised the text to correct this issue.

3. Locations of the observatories referred should be shown on the map. This could be included in Figure 8.

   We have included a new figure with a map showing the location of all observatories used in this work.

4. Figure 4: The time variation is unclear. The vertical scale should be changed so that the variation is clearer.

   The referee is right. We have modified this figure by removing the values of the *gufm1* model. As the first referee pointed out, the *gufm1* model cannot be used to show daily values. Then, the vertical scale of the figure has been adjusted to show daily variations more clearly.

5. lines 24 and 31:What "Supplementary Material" do refer to?

   Thank you for pointing out this issue. The specific section of the Supplementary Material containing this information has now been provided.

6. *Lines 25-26: "that this observatory does not have a great tradition on geomagnetism"): What does this mean? Is RMO not important?*

   The Astronomical Observatory (ROM) does not have a long tradition in geomagnetism. Regular measurements of magnetic declination and inclination were only made between 1879 and 1901, and it was never equipped with variometers for continuous registration of the field. We have modified the text to make it clearer.

7. *Figure 2 : Zero declination horizontal line should be indicated.*

   Following this suggestion, we have highlighted the zero-declination line in this and other figures for better interpretation.

8. *A brief explanation of the geomagnetic database (HISTMAG, gufmq1model), is necessary*

   The text already includes a brief introduction to the HISTMAG database (lines 260-264) and to the gufm1 model (lines 47-48), as well as references to the published articles. We believe that these explanations are sufficient for the purpose of our work.

9. Is it possible to know the date and location of the first self-recording magnetometer observations?

   The Royal Observatory of Madrid never was equipped with variometers. In Spain, the first observatory equipped with variometers was San Fernando Observatory. A set of Adie variometers was purchased in 1874, but the San Fernando yearbooks began to be published in 1891.